# Factor XIII Activity Might Already Be Impaired before Veno-Venous ECMO in ARDS Patients: A Prospective, Observational Single-Center Cohort Study

**DOI:** 10.3390/jcm10061203

**Published:** 2021-03-14

**Authors:** Onnen Moerer, Jan Felix Huber-Petersen, Joern Schaeper, Claudia Binder, Saskia Wand

**Affiliations:** 1Department of Anesthesiology, University Medical Center Goettingen, 37075 Goettingen, Germany; felix-hp@gmx.de (J.F.H.-P.); joern.schaeper@med.uni-goettingen.de (J.S.); saskia.wand@med.uni-goettingen.de (S.W.); 2Department of Hematology and Oncology, University Medical Center Goettingen, Robert-Koch-Str. 40, 37075 Goettingen, Germany; claudia.binder@med.uni-goettingen.de; 3Department of Anesthesiology, Ruhr-University Bochum, St. Josef- and St. Elisabeth Hospital, 44791 Bochum, Germany

**Keywords:** extracorporeal membrane oxygenation, ECMO, ARDS, coagulation, factor XIII

## Abstract

Direct complications in patients receiving extracorporeal (veno-venous) membrane oxygenation (vvECMO) are mainly either due to bleeding or thromboembolism. We aimed to evaluate the course of routine coagulation parameters and the activity of different coagulation factors—with special focus on factor XIII (F XIII)—before, during and after vvECMO in acute respiratory distress syndrome (ARDS) patients. The activity of coagulation factors and rotational thrombelastometry were analyzed in 20 ECMO patients before (T-1) and 6 h (T0), one (T1), three (T3) and seven days (T7) after the implantation, as well as one and three days after the termination of ECMO. F XIII activity was already severely decreased to 37% (30/49) before ECMO. F XIII activity was the only coagulation factor continuously declining during vvECMO, being significantly decreased at T3 (31% (26/45) vs. 24% (18/42), *p* = 0.0079) and T7 (31% (26/45) vs. 23% (17/37), *p* = 0.0037) compared to T0. Three days after termination of vvECMO, platelet count and fibrinogen nearly doubled and factors II, V, XI and XIII showed spontaneous significant increases. Severe ARDS patients showed a considerably diminished factor XIII activity before vvECMO initiation and its activity continuously declined later on. Thus, incorporation of F XIII monitoring into the regular hemostaseologic routine during vvECMO therapy seems advisable. Due to the potential development of a hypercoagulatory state after the termination of vvECMO, tight hemostasiologic monitoring should persist in the initial phase after ECMO termination.

## 1. Introduction

Extracorporeal Membrane Oxygenation (ECMO) has evolved to a widely applied rescue therapy for patients with severe acute hypoxemic respiratory failure, which is refractory to conventional therapy [1,2,3]. However, despite its potentially lifesaving role in patients with severe acute respiratory distress syndrome (ARDS), the therapy itself has been related to severe complications, among which are thromboembolic or hemorrhagic events, which are frequently observed [4,5,6,7,8,9,10].

Usually, systemic anticoagulation therapy is unavoidable as a therapeutic measure in order to prevent clot formation during extracorporeal circulation [1,2,3]. This adds an additional risk of bleeding, since the coagulation system of the critically ill might be already impaired before the initiation of ECMO due to the underlying disease [11]. The extracorporeal circulation can cause further alterations such as thrombocytopenia, temporary platelet dysfunction, acquired von Willebrand syndrome, as well as factor I and factor XIII (F XIII) deficiencies, which have been described during extracorporeal (veno-venous) membrane oxygenation (vvECMO) therapy [9,12,13,14]. Bleeding events during the course of treatment are frequent and might lead to premature termination of the technique or have a critical impact on outcome [15,16].

Point-of-care (POC) based algorithms for the management of acute bleeding events can reduce transfusion rates in patients [17]. Nevertheless, during ECMO therapy POC methods alone seem not to be able to predict bleeding events sufficiently [18]. An early comprehensive analysis including pro- and anticoagulatory factors as well as POC parameters could enable the detection of the impairment of the coagulation system prior to initiation and during the initial course of ECMO therapy. This might help to avoid thromboembolic and bleeding complications later on.

In particular, F XIII can only be indirectly assessed by POC methods. An acquired F XIII deficiency can be associated with major complications such as surgical re-exploration in critically ill patients, putting the need for a routine monitoring of F XIII activity in intensive care medicine more in focus [19,20,21].

The aim of this observational study was to evaluate the time course of the activity of different coagulation factors, with special focus on F XIII, as well as routine and POC coagulation parameters before, during and after vvECMO therapy in patients with severe ARDS.

## 2. Materials and Methods

This prospective, observational monocentric cohort study was performed at the University Medical Center Goettingen, Germany. Part of the study data have been presented to national and international congresses. Data on platelet function using multiple electrode aggregometry (MEA, Multiplate) have been previously published [12]. The study was registered at German Clinical Trials Register (DRKS00005650), complies with the declaration of Helsinki and was approved by the local Scientific and Ethics Review Board (19/5/13).

### 2.1. Study Population

All adult patients with the need for a vvECMO treatment were screened for eligibility.

The indication for vvECMO treatment was set by the responsible physician of the ECMO team. Patients were either placed on ECMO in the referral hospital and retrieved on ECMO or transported to the ARDS center and eventually placed on ECMO, if conservative treatment failed. Venous access was established percutaneously using the right femoral vein for blood outflow (21–25 Fr lumen Cannulae, Maquet Critical Care, Solna, Sweden) and the right internal jugular vein (17–23 Fr) for return flow. vvECMO was established with conventional systems (ROTAFLOW^®^ or CARDIOHELP^®^ system with Quadrox-D^®^ oxygenator/HLS/PLS, Maquet Critical Care, Solna, Sweden), and the ECMO circuit was primed with saline and connected via a biocoated tubing set. Unfractionated heparin was used as anticoagulant with continuous intravenous application targeting an activated clotting time (ACT) of > 140.

Due to the nature of the disease (severe ARDS with the indication for ECMO as a rescue measure), the fact that none of the patients were conscious at the time of enrollment and that ECMO was not to be delayed, the ethics committee agreed to the following procedure.

Patients were initially included based on presumed will, which was assessed via the treating clinician if the relatives could not be accessed and official consent/denial was obtained thereafter. Secondarily consent was obtained from the appointed legal guardian. In case of refusal, data collection was stopped, and the patient was excluded from the study. After full medical recovery, written informed consent was additionally obtained from the patients.

A known disorder in the coagulation system or pregnancy were exclusion criteria for enrollment in the study.

### 2.2. Blood Samples and Laboratory Analysis

Conventional laboratory analyses, including hemoglobin, leucocyte and platelet counts, prothrombin time (PT), activated partial thromboplastin time (aPTT), fibrinogen (Clauss assay) and Antithrombin (AT), were conducted during the daily routine and an ACL TOP 750 hemostasis testing system using the respective Hemosil^®^ reagents (Instrumentation Laboratory, Werfen GmbH, Munich, Germany) was used for analysis. Blood samples for the measurement of the activity of coagulation factors were drawn before (T-1) and 6 h (T0), one (T1), three (T3) and seven days (T7) after the implantation of vvECMO as well as one (post-ECMO T1) and three days (post-ECMO T3) after the termination of ECMO treatment. Factor activities were determined on the ACL TOP system using the respective deficient plasma together with the ReadiPlasTin^®^ or SynthASil^®^ reagents (all Hemosil, Werfen GmbH, Munich, Germany). F XIII measurements were performed with the Hemosil F XIII antigen (Werfen GmbH, Munich, Germany). Blood samples were processed within two hours by the hospital’s Department of Hemostaseology. For rotational thrombelastometry, blood samples were taken before (T-1) and 6 h (T0), one (T1), two (T2), three (T3) and seven days (T7) after the initiation of vvECMO treatment.

### 2.3. Rotational Thrombelastometry

A ROTEM^®^ device was used for rotational thrombelastometry (TEM International GmbH, Munich, Germany). All tests were performed according to the manufacturer’s instructions, using 300 µL citrated blood for each test. The tissue factor activated EXTEM test, the contact activated INTEM test, the FIBTEM test, which assesses the fibrinogen part of coagulation by blocking platelet activity with Cytochalasin D in a tissue factor activated test, and the HEPTEM test, which combines contact activation with heparinase to detect possible effects of heparin, were performed at each timepoint.

### 2.4. Sample Size and Statistics

Due to the exploratory nature of the study, no sample size calculation was performed. Analysis for normal distribution was performed using the Kolmogorov–Smirnov test. Data are described as median with interquartile range (IQR) or mean ± standard error of the mean (SEM) accordingly. Friedman repeated measures analyses of variance on ranks were used to assess differences between timepoints. Specific posthoc tests were performed when appropriate. *p* values were adjusted for multiple testing using the Bonferroni method.

Differences between groups were analyzed using the Mann–Whitney test or the Fisher exact test as appropriate.

All statistical analyses were performed using Statistica 13.3 (Tibco Software Inc., Palo Alto, CA, USA).

## 3. Results

A total of *n* = 31 patients with severe ARDS and the indication vvECMO as a rescue therapy were assessed for eligibility. Of these, *n* = 1 fulfilled the exclusion criteria, *n* = 1 refused to participate, *n* = 3 already participated in other studies and *n* = 6 patients could not be enrolled due to logistical reasons. Twenty patients (median age 57, IQR 52/64), 13 (65%) of whom were male, were finally included in the study. Median ICU stay was 19 days (IQR 13.5/51), of which 9.5 days (IQR 7/13.5) were spent in ECMO. Twenty-eight-day mortality was 45%; 11 out of 20 patients died in the ICU (ICU mortality—55%).

During the observational period of seven days, seven patients (35%) showed signs of mild bleeding, and one patient (5%) suffered from an intracerebral bleeding on the second day of ECMO treatment. This patient had a known cerebral ischemia on enrollment and ECMO treatment was initiated with awareness of the increased risk for an intracerebral bleeding event. Two patients out of 20 required an oxygenator or ECMO circuit change, respectively, due to thromboembolic complications.

Substitution of prothrombin complex concentrate (PPSB) and fibrinogen was necessary in *n* = 4 and *n* = 2 patients, respectively, during the observational period. A transfusion of platelet concentrates occurred in *n* = 3 patients. These substitutions were in accordance with the hemotherapy algorithms of our institution. Additionally, *n* = 4 patients received F XIII concentrate between day 4 and day 7. None of the patients were substituted with fresh frozen plasma during the observational period.

### 3.1. Standard Laboratory Data

The initiation of vvECMO treatment led to a significant decrease in hemoglobin (10.1 (8.6/13) vs. 8.85 mg/dL (8.25/10.1), *p* = 0.0063), platelet count (145 (128/210) vs. 126/nL (94/188), *p* = 0.0029) and leucocyte count (16.77 (8.7/21.3) vs. 12.2/nL (6.7/16.2), *p* = 0.0058) between T-1 and T0. The platelet count decreased continuously throughout the observational period, with a significant difference between T0 and T7 (126 (94/188) vs. 102/nL (65/135), *p* = 0.0056). In contrast, AT levels showed a significant increase at T3 and T7 when compared with AT levels at T0 (43% (37/66) vs. 59% (47/84), *p* = 0.0009 and 43% (37/66) vs. 85% (60/103), *p* = 0.0008).

The platelet count three days after decannulation was significantly higher than the last value measured during ECMO treatment (95 (44/151) vs. 232/nL (139/331), *p* = 0.0125). Additionally, fibrinogen levels nearly doubled between these two timepoints (248 (207/527) vs. 491 mg/dL (285/636), *p* = 0.0077).

The complete routine laboratory data during vvECMO therapy and after the termination of extracorporeal circulation are given in Appendix A, respectively.

### 3.2. Rotational Thrombelastometry

Thrombelastometry parameters were not markedly influenced six hours after the initiation of ECMO treatment. With increasing duration of ECMO therapy, the clotting time (CT) in the INTEM and the clot formation time (CFT) in the INTEM and EXTEM showed significant increases, while the A 10 in the EXTEM and the maximum clot firmness (MCF) in the EXTEM and INTEM significantly decreased (Appendix A). Nevertheless, the MCF in both assays stayed within the normal range during the observational period and the CT in the EXTEM was only slightly elevated at T7. The CT in the FIBTEM assay showed a significant increase at T7 (67 (60/78) vs. 83 (71/96), *p* = 0.0026). However, the A 10 and the MCF in the FIBTEM assay displayed an increasing tendency throughout the observational period with a median MCF slightly above the normal range at T3. Viscoelastic variables in patients who experienced bleeding events did not differ from measurements in patients without signs of bleeding during the observational period.

### 3.3. Activity of Coagulation Factors

Six hours after the initiation of ECMO therapy, all measured coagulation factors showed a reduction ranging between 7% and 26% (Table 1). During the further course of ECMO treatment, the level of all coagulation factors listed in Table 1 stayed the same or displayed an increasing tendency. In contrast, the activity of F XIII was significantly lower at T3 (31% (26/45) vs. 24% (18/42), *p* = 0.0079) and T7 (31% (26/45) vs. 23% (17/37), *p* = 0.0037) in comparison with T0 (Figure 1).

Three days after the termination of ECMO treatment, the activity of all measured coagulation factors was higher than values at T0. The activity of F XIII spontaneously increased three days after the termination of ECMO therapy compared to the last value measured during ECMO (28% (20/48) vs. 46% (41/59), *p* = 0.02) (Figure 2). This significant increase was also observed for the activity of factors II, V and XI (Table 2).

No significant differences in the activity of coagulation factors in patients were detected with respect to a bleeding event during the observational period.

## 4. Discussion

The present observational study evaluated the activity of different coagulation factors, as well as routine and POC coagulation parameters during vvECMO therapy. Neither the implantation nor the following course of vvECMO therapy remarkably altered the variables of rotational thrombelastometry. Though the mean CT in the INTEM and EXTEM slowly increased over time and the mean MCF slowly decreased, the results mostly remained within or close to the reference range for the respective parameters during the observational period. Rotational thrombelastometry does not seem to have a predictive value with respect to bleeding events during vvECMO therapy [18]. Therefore, monitoring patients during vvECMO therapy with rotational thrombelastometry on a regular basis does not seem useful, since the method cannot depict many of the acquired coagulation deficiencies during vvECMO therapy [13].

The activity of all coagulation factors but factor XIII was within the normal reference range before the initiation of vvECMO therapy. The initiation of vvECMO therapy led to a reduction in the activity of all coagulation factors measured in the present study. This effect could be either due to the hemodilution at the beginning of vvECMO therapy or related to consumption of coagulation factors by contact activation. However, the kinetics of recovery during the observational period differed considerably between the coagulation factors. In particular, coagulation factors with shorter biological half-lives, such as F V, F VII, F VIII, F IX and F X, showed continuously increasing activity, almost all reaching or even surpassing the baseline activity before vvECMO at day 7. In contrast, the coagulation factors with longer biological half-lives, such as F II, F XI, F XII and F XIII, were not able to recover their activity during this time. These dynamics have been partially described in cardiac patients receiving vaECMO [22,23]. In particular, the reduced activity of F XII might even prove beneficial for patients with extracorporeal circulation, since an inhibited F XII prevented fibrin deposition and thrombosis in the extracorporeal circuit in an animal model [24,25].

Interestingly, the activity of F XIII showed a completely different behavior during vvECMO compared to all other coagulation factors evaluated in the present study. Acquired F XIII deficiency in patients with vvECMO support has been described before [13]. The present study supports that low levels of F XIII in patients receiving vvECMO are already present hours before the initiation of vvECMO therapy [26]. Furthermore, the results of the present study indicate that the kinetics of F XIII activity under vvECMO therapy differ completely from all other evaluated coagulation factors. F XIII activity decreased continuously during the complete observational period and reached critically low levels, while all other coagulation factors displayed stable or even increasing activity within or slightly below the normal reference range. One reason for this decreasing activity could be consumption of F XIII due to activation of the coagulation system by extracorporeal circulation. Additionally, F XIII has the longest biological half-life of all coagulation factors with 120–200 h. This results in slower secretion kinetics, which may impede the compensation of sudden decreases in activity. F XIII is a known link between coagulation and the immune system [27,28]. Low levels of F XIII have been described for patients suffering from severe sepsis or septic shock [29]. Results from animal models even suggest that crosslinking of fibrin monomers through F XIII might promote the formation of microemboli and consecutive organ damage in severe sepsis [30]. The number of patients in the present study is too small to evaluate potential predictive values of the activity of F XIII with respect to bleeding complications or outcome in patients with vvECMO. Prospective studies are needed to define target values of activity and further examine the role of F XIII in the management of patients receiving vvECMO therapy.

There is lack of data on the therapeutic consequence of the detection of low F XIII activity in patients without suffering from bleeding. According to the ELSO guidelines, “fresh frozen plasma or specific clotting factors may be indicated if deficiencies are demonstrated” in patients with bleeding complications after returning coagulation status to normal “as much as possible” [31]. However, F XIII activity is only increased by about 1%–2% if 1 mL per kg body weight fresh frozen plasma is transfused. Taking into account the increased risk and the low baseline levels in our patients, substitution of F XIII concentrate might be more reasonable. According to a national guideline (therapy with blood components and plasma derivate), factor concentrates instead of FFP should be used if deficits are more efficiently and compatible corrected [32]; this is especially important since the deficit difference of the different coagulation factors would lead to oversubstitution if F XIII is normalized by plasma concentrates. While others suggest targeting activity levels of >50% [13], we rather aimed at an initial normalization of values and accepted subnormal values later on if there was no sign of broader deterioration of coagulation and no signs of bleeding, which was accompanied by reassessment. In addition to the coagulation perspective, the potential impact on inflammation and wound healing of F XIII might prove interesting in the future [33,34].

ECMO therapy is a constant navigation between hemorrhagic and thromboembolic complications. Our results demonstrate, that pro- and anticoagulatory factors change dynamically during extracorporeal circulation. Hence, a monitoring of the activity of selected coagulation factors on a regular basis could represent a useful component in the hemostaseologic management of patients receiving vvECMO therapy.

The significant spontaneous increase in F II, F V, F XI and F XIII, as well as the increase in platelet counts and fibrinogen levels 72 h after the termination of vvECMO as seen in the present study, illustrate that the dynamic changes in the coagulatory system do not stop with the termination of vvECMO. We also observed a significant increase in AT in this phase. Nevertheless, considering the relatively high incidence of thrombotic complications after vvECMO therapy, there seems to be the need for a more complex hemostaseologic monitoring and perhaps a more intensified anticoagulation in the phase after the termination of extracorporeal circulation as well [35,36,37,38].

Our study has several limitations. First, the small sample size analyzed only allows our results to be generalized to a limited extent. Second, due to the unblinded, observational design of the study, the results of the coagulation factor activity test were accessible for the attending physicians. This led to a substitution of F XIII in four patients after day 4 of the observational period due to critically low levels not due to acute bleeding complications. Taking this increased awareness into account, the F XIII time course would presumably be even more impressive if blinded and untreated in these four patients. To minimize the effect of this substitution on our results, we decided to omit all F XIII activity measurements after a patient received an application of F XIII concentrate.

Finally, the focus of this study was mainly on procoagulatory factors. AT was the only anticoagulatory factor analyzed. Considering the observed changes in the phase after termination of vvECMO, the measurement of Protein C could have been an interesting complementation to the results of coagulation factor activity.

## 5. Conclusions

In conclusion, the activity of F XIII in patients receiving vvECMO therapy is already considerably diminished before the initiation of extracorporeal circulation. During vvECMO therapy, F XIII is the only coagulation factor displaying a continuously declining activity. Therefore, we recommend incorporating the monitoring of F XIII activity into the regular hemostaseologic routine during vvECMO therapy. There is increasing evidence that patients may develop a hypercoagulatory state after the termination of vvECMO with an increased risk of thrombotic complications. The present study demonstrates that the activity of different coagulation factors as well as platelet count and fibrinogen levels significantly increase within 72 h after decannulation. Amongst other reasons, this might contribute to an increased risk of thrombosis after vvECMO therapy. Further studies are needed to determine the duration of this state and the implications for anticoagulation guidelines in the days following vvECMO therapy.

## Figures and Tables

**Figure 1 jcm-10-01203-f001:**
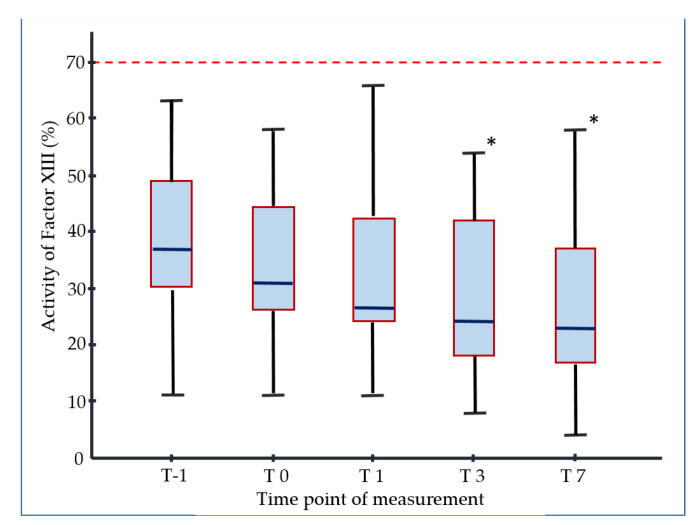
Activity of F XIII before (T-1) and after the initiation of vvECMO therapy over an observational period of seven days: 6 h (T0), 1 day (T1), 3 days (T2) and 7 days (T7); * indicates significant decrease in F XIII activity compared to T0; dotted line represents threshold to the normal reference range of F XIII.

**Figure 2 jcm-10-01203-f002:**
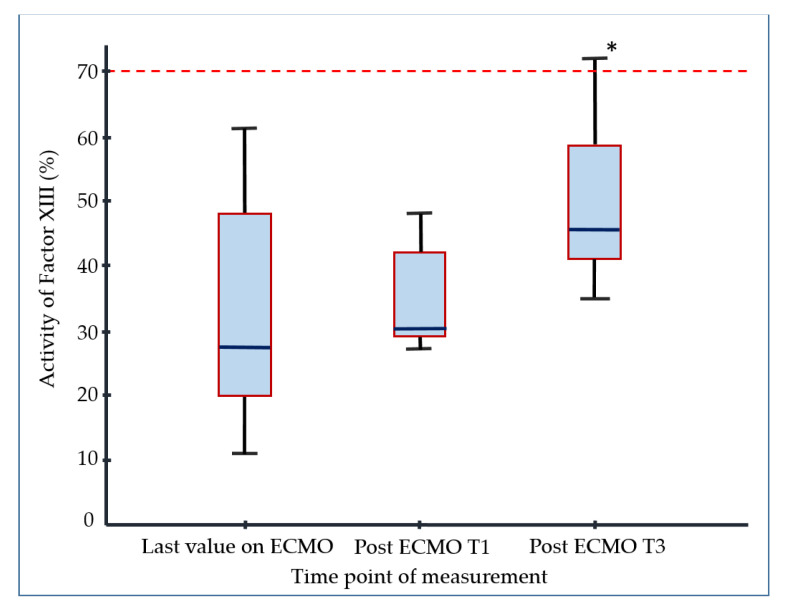
Activity of F XIII before ECMO termination (last value ECMO; *n* = 14) and 1 (post-ECMO T1; *n* = 7) as well as 3 days (post-ECMO T3; *n* = 8) after the termination of vvECMO therapy; * indicates significant increase in F XIII activity compared to last value ECMO; dotted line represents threshold to the normal reference range of F XIII.

**Table 1 jcm-10-01203-t001:** Activity of different coagulation factors (excluding factor XIII (F XIII)) in acute respiratory distress syndrome (ARDS) patients before and after the initiation of veno-venous extracorporeal membrane oxygenation (vvECMO).

	Coagulation Factor Activity (%) at Different Timepoints				
Factor	T-1 (*n* = 11)	T0 (*n* = 20)	T1 (*n* = 20)	T3 (*n* = 19)	T7 (*n* = 15)
**Factor II**	73 (68/78)	55 (42/60)	56 (46/75)	68 (47/71)	59 (43/71)
**Factor V**	132 (129/145)	101 (59/131)	106 (56/140)	98 (81/145)	110 (92/150)
**Factor VII**	62 (53/75)	53 (43/76)	60 (36/87)	64 (55/97)	69 (52/93)
**Factor VIII**	200 (185/200)	187 (156/200)	200 (161/200)	200 (142/200)	178 (142/200)
**Factor IX**	150 (126/150)	122 (100/150)	138 (109/150)	141 (122/150)	148 (134/150)
**Factor X**	73 (56/103)	56 (47/82)	61 (47/96)	66 (56/91)	77 (61/89)
**Factor XI**	101 (99/127)	72 (55/97)	73 (62/101)	68 (61/102)	77 (57/101)
**Factor XII**	82 (52/88)	51 (38/60)	52 (41/67)	52 (41/67)	48 (39/66)

**Table 2 jcm-10-01203-t002:** Coagulation factor activity (%) before and after termination of ECMO therapy.

	Before Termination of ECMO (*n* = 10)	Post-ECMO Day 1 (*n* = 10)	Post-ECMO Day 3 (*n* = 10)
**Factor II**	69 (45/75)	74 (71/82)	* 77 (57/95)
**Factor V**	112 (92/140)	125 (86/149)	* 134 (112/150)
**Factor VII**	69 (50/87)	67 (57/84)	62 (47/101)
**Factor VIII**	200 (148/200)	200 (180/200)	200 (200/200)
**Factor IX**	141 (130/150)	148 (132/150)	150 (140/150)
**Factor X**	78 (55/97)	88 (71/108)	84 (71/108)
**Factor XI**	95 (74/102)	118 (96/134)	* 144 (101/148)
**Factor XII**	60 (39/75)	69 (46/95)	63 (50/105)

* significant difference compared to the last value before termination of vvECMO therapy; all values are given as median with interquartile range.

## Data Availability

The datasets used and analyzed for the current study are available from the corresponding author on reasonable request.

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
