# Peer review of "Factor XIII Activity Might Already Be Impaired before Veno-Venous ECMO in ARDS Patients: A Prospective, Observational Single-Center Cohort Study"

_jcm, 2021, doi:10.3390/jcm10061203_

Round 1
Reviewer 1 Report
brief summary
This observational study examines the different coagulation factors during V-V ECMO support. The article focusses on factor XIII as this is the only coagulation factor that is diminished before the initiation of ECMO and its activity is further decreasing during support.
broad and specific comments
The reviewed article has an interesting and clinically relevant subject, because thrombosis and bleeding during ECMO support remain an important issue and especially because factor XIII activity is not detected by conventional laboratory parameters. As stated by the authors there is however no proven relationship of impaired factor XIII activity with clinically relevant bleeding yet. Therefore we have the following remaining questions and minor remarks;
- there is discrepancy between the statement of incorporation of F XIII monitoring during VV ECMO support in the abstract compared to the conclusion: seems advisable (line 28) vs strongly recommended (line 280). The latter term seems disproportionate and should at least be nuanced.
- part of the study data has allready been published in J. Clin. Med. -> change published elsewhere [12] (line 68) in has been previously published
- substitution of PPSB and fibrinogen as well as F XIII is mentioned (lines 139-142) but there is no statement about possible transfusion with fresh frozen/omni plasma. We would also like to know if there were (several) changes of oxygenator, as this could influence coagulation as well. The same goes for the level of anticoagulation (with heparin we presume)
- the level of all coagulation factors but one (line 175) -> is this one factor, factor XIII? Because factor I (fibrinogen) is also diminishing (although not significantly according to the authors)
- perhaps the authors could highlight a bit more about the close interaction of factor XIII with the immune system; eg. were there other inflammatory parameters measured during this observational study?
- dare the authors to give advise about the way we should correct critically low levels of factor XIII? Simply by giving plasma or isolated factor XIII?
Author Response
Please see attachement

Reviewer 2 Report
Authors observed factor XIII level was decreased in patients with ARDS, and progressive decrease during the course of ECMO. It is an interesting topic, however, unfortunately the manuscript was not well written. Following is my criticism.
- There was no mentioning of assay methods for each factor including factor XIII.
- When ECMO was initiated, was the circuit primed with saline, red cells, or red cells and plasma?
- What kind of anticoagulant was used, heparin or other direct thrombin inhibitor?
- Why did 4 patients receive factor XIII? Was it factor XIII concentrate or recombinant factor XIII? Since they removed from the analysis, was the total n-16?
- Was there bleeding symptoms in patients with a low factor XIII? After giving factor XIII, was bleeding stopped?
- There is no information on transfusion with plasma and platelets? They should have affected the factor levels.
- Most importantly, I need to see controls; factor XIII levels of ARDS without vvECMO or factor XIII levels in vaECMO.
- Post ECMO T3, factor XIII level was increased, but still lower than the reference range. Was it ever normalized?
Author Response
Dear Reviewer,
we are thankful for your comments which let to adaptations of our manuscript. Although we were not able to transfer all the suggestions, we sincerely hope that the you agree with our improvements. However, if there are further comments, we are definitively willing to address these issues.
With kind regards
Onnen Moerer
Point-by-point response to the reviewer’s comments
REVIEWER 2
- There was no mentioning of assay methods for each factor including factor XIII.
Comment: We appreciate the reviewer’s suggestions and added the information to the methods section.
- When ECMO was initiated, was the circuit primed with saline, red cells, or red cells and plasma?
Comment: We added the information, that we routinely prime the circuit with saline, to the methods section.
- What kind of anticoagulant was used, heparin or other direct thrombin inhibitor?
Comment: Thanks for requesting this information. We routinely use heparin and have added this information to the methods section.
- Why did 4 patients receive factor XIII? Was it factor XIII concentrate or recombinant factor XIII? Since they removed from the analysis, was the total n-16?
Comment: When the study was conducted, routine analysis and supplementation of Factor XIII was not a given standard at our ICU, although the team was aware of the problem. Since we could not fully blind the team for the lab observation for ethical reasons, a reaction to the results could not be completely omitted. Thus in 4 patients F XIII concentrate was given. Since no patients were substituted before T3, we included the results of all patients up to T3. The n-number decreased during the observational period due to either termination of ECMO, death or because of FXIII substitution. This limitation of the study data is addressed in the limitations section. We have to conclude that the observed deficit would have been even higher without the substitution.
- Was there bleeding symptoms in patients with a low factor XIII? After giving factor XIII, was bleeding stopped?
As mentioned above, FXIII concentrate was not substituted due to acute bleeding complications but rather due to very low measured FXIII levels. As far as it can be assessed by the results, Factor XIII was not a predictor for bleeding complications. For a better illustration we provide the following table. However, I would be careful in the interpretation of this data since the number of bleeding complications was too low to allow for serious statistics.
|
Signs of bleeding on ECMO |
T -1 (n=9) |
T 0 (n=20) |
T 1 (n=20) |
T 3 (n=19) |
T 7 (n=12) |
FXIII activity [%] |
Yes (n=8) |
48 (37/49) (n=5) |
39 (28/45) (n=8) |
32 (25/39) (n=8) |
27 (19/41) (n=8) |
19 (13/44) (n=5) |
No (n=12) |
29 (19/41) (n=4) |
30 (26/42) (n=12) |
27 (25/46) (n=12) |
24 (16/42) (n=11) |
25 (21/30) (n=7) |
- There is no information on transfusion with plasma and platelets? They should have affected the factor levels.
Comment: The results section has been revised accordingly and information on transfusion of plasma and platelets has been added.
- Most importantly, I need to see controls; factor XIII levels of ARDS without vvECMO or factor XIII levels in vaECMO.
Comment:
We agree that comparing FXIII levels of ECMO-ARDS patients with a non ECMO-ARDS control would be very interesting. We hoped to spark exactly these kind of ideas with this observational study for further investigation of this interesting field in the treatment of ARDS patients. With regard to the interest of the reviewer in seeing controls to the present data however, we have to admit that we are not able to expend this study to a control group for several reasons. As recognized, there was a large study protocol, which included a repeated full set of coagulation factor analysis, conventional and point of care diagnostics. Due to the high cost for the analysis per patient, it would not have been feasible for us to include another non-ECMO group into this study.
Additionally, based on our findings we would rather recommend a study design with a smaller diagnostic setup only targeting a core set of laboratory variables, if aiming for a comparison between non-ECMO and ECMO ARDS patients. Planning such a study taking different levels of severance of ARDS into consideration would be an interesting future project and we would sincerely be pleased if the reviewer would join us for such a study, which essentially would probably not request a high number of patients but probably 7-10 each per group (mild – moderate – severe ARDS plus ARDS with ECMO as a rescue therapy)
However, based on our study, we would be even more interested in conducting an RCT comparing a target based early substitution with standard of care.
- Post ECMO T3, factor XIII level was increased, but still lower than the reference range. Was it ever normalized?
Comment: There were patients where we saw normalized values. However, the study protocol ended with the described measurements three days after the termination of ECMO. Thus, further F XIII analysis were only indicated on an irregular basis and not in all patients. It is hard to draw conclusions from these measurements.
Round 2
Reviewer 2 Report
Authors answered my comments appropriately. I would accept with minor change below.
Line 155: thromboembolic complication --> clot formation
Author Response
Dear Reviewer,
there was a minor comment (Line 155: thromboembolic complication --> clot formation) that we have changed accordingly.
Best regards
Onnen Moerer